# Intrinsic-Extrinsic Convolution and Pooling for Learning on 3D Protein Structures

**Pedro Hermosilla**
Ulm University

**Marco Schäfer**
Tübingen University

**Matěj Lang**
Masaryk University

**Gloria Fackelmann**
Ulm University

**Pere Pau Vázquez**
University of Catalonia

**Barbora Kozlíková**
Masaryk University

**Michael Krone**
Tübingen University

**Tobias Ritschel**
University College London

**Timo Ropinski**
Ulm University

## Abstract

Proteins perform a large variety of functions in living organisms and thus play a key role in biology. However, commonly used algorithms in protein learning were not specifically designed for protein data, and are therefore not able to capture all relevant structural levels of a protein during learning. To fill this gap, we propose two new learning operators, specifically designed to process protein structures. First, we introduce a novel convolution operator that considers the primary, secondary, and tertiary structure of a protein by using $n$-D convolutions defined on both the Euclidean distance, as well as multiple geodesic distances between the atoms in a multi-graph. Second, we introduce a set of hierarchical pooling operators that enable multi-scale protein analysis. We further evaluate the accuracy of our algorithms on common downstream tasks, where we outperform state-of-the-art protein learning algorithms.

## 1 Introduction

Proteins perform specific biological functions essential for all living organisms and hence play a key role when investigating the most fundamental questions in the life sciences. These biomolecules are composed of one or several chains of amino acids, which fold into specific conformations to enable various biological functionalities. Proteins can be defined using a

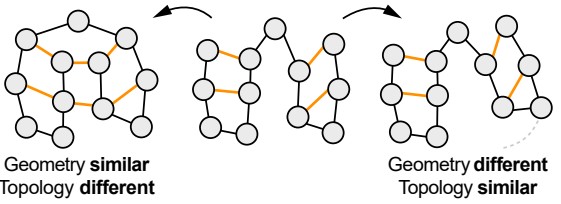

Geometry **similar** Topology **different**     Geometry **different** Topology **similar**

Figure 1: Invariances present in protein structures.

multi-level structure:: The *primary* structure is given by the sequence of amino acids that are connected through covalent bonds and form the protein backbone. Hydrogen bonds between distant amino acids in the chain form the *secondary* structure, which defines substructures such as $\alpha$-helices and $\beta$-sheets. The *tertiary* structure results from protein folding and expresses the 3D spatial arrangement of the secondary structures. Lastly, the *quarternary* structure is given by the interaction of multiple amino acid chains.

Considering only one subset of these levels can lead to misinterpretations due to ambiguities. As shown by Alexander et al. (2009), proteins with almost identical primary structure, i.e., only containing a few different amino acids, can fold into entirely different conformations. Conversely, proteins from SH3 and OB folds have similar tertiary structures, but their primary and secondary structures differ significantly (Agrawal & Kishan, 2001) (Fig. 1). To avoid misinterpretations arising from these observations, capturing the invariances with respect to primary, secondary, and tertiary structures is of key importance when studying proteins and their functions.

Previously, the SOTA was dominated by methods based on hand-crafted features, usually extracted from multi-sequence alignment tools (Altschul et al., 1990) or annotated databases (El-Gebali et al., 2019). In recent years, these have been outperformed by protein learning algorithms in different

protein modeling tasks such as protein fold classification (Hou et al., 2018; Rao et al., 2019; Bepler & Berger, 2019; Alley et al., 2019; Min et al., 2020) or protein function prediction (Strodthoff et al., 2020; Gligorijevic et al., 2019; Kulmanov et al., 2017; Kulmanov & Hoehndorf, 2019; Amidi et al., 2017). This can be attributed to the ability of machine learning algorithms to learn meaningful representations of proteins directly from the raw data. However, most of these techniques only consider a subset of the relevant structural levels of proteins and thus can only create a representation from partial information. For instance, due to the high amount of available protein sequence data, most techniques solely rely on protein sequence data as input, and apply learning algorithms from the field of natural language processing (Rao et al., 2019; Alley et al., 2019; Min et al., 2020; Strodthoff et al., 2020), 1D convolutional neural networks (Kulmanov et al., 2017; Kulmanov & Hoehndorf, 2019), or use structural information during training (Bepler & Berger, 2019). Other methods have solely used 3D atomic coordinates as an input, and applied 3D convolutional neural networks (3DCNN) (Amidi et al., 2017; Derevyanko et al., 2018) or graph convolutional neural networks (GCNN) (Kipf & Welling, 2017). While few attempts have been made to consider more than one structural level of proteins in the network architecture (Gligorijevic et al., 2019), none of these hybrid methods incorporate all structural levels of proteins simultaneously. In contrast, a common approach is to process one structural level with the network architecture and the others indirectly as input features(Baldassarre et al. (2020) or Hou et al. (2018)).

In this paper, we introduce a novel end-to-end protein learning algorithm, that is able to explicitly incorporate the multi-level structure of proteins and captures the resulting different invariances. We show how a multi-graph data structure can represent the primary and secondary structures effectively by considering covalent and hydrogen bonds, while the tertiary structure can be represented by the spatial 3D coordinates of the atoms (Sec. 3). By borrowing terminology from differential geometry of surfaces, we define a new convolution operator that uses both intrinsic (primary and secondary structures) and extrinsic (tertiary and quaternary structures) distances (Sec. 4). Moreover, since protein sizes range from less than one hundred to tens of thousands of amino acids (Brocchieri & Karlin, 2005), we propose protein-specific pooling operations that allow hierarchical grouping of such a wide range of sizes, enabling the detection of features at different scales (Sec. 5). Lastly, we demonstrate, that by considering all mentioned protein structure levels, we can significantly outperform recent SOTA methods on protein tasks, such as protein fold and enzyme classification.

Code and data of our approach is available at `https://github.com/phermosilla/IEConv_proteins`.

## 2 RELATED WORK

Early works on learning protein representations (Asgari & Mofrad, 2015; Yang et al., 2018) used word embedding algorithms (Mikolov et al., 2013), as employed in Natural Language Processing (NLP). Other approaches have used 1D convolutional neural networks (CNN) to learn protein representations directly from an amino acid sequence, for tasks such as protein function prediction (Kulmanov et al., 2017; Kulmanov & Hoehndorf, 2019), protein-compound interaction (Tsubaki et al., 2018), or protein fold classification (Hou et al., 2018). Recently, researchers have applied complex NLP models trained unsupervised on millions of unlabeled protein sequences and fine-tune them for different downstream tasks (Rao et al., 2019; Alley et al., 2019; Min et al., 2020; Strodthoff et al., 2020; Bepler & Berger, 2019). While representing proteins as amino acid sequences during learning, is helpful when only sequence data is available, it does not leverage the full potential of spatial protein representations that become more and more available with modern imaging and reconstruction techniques.

To learn beyond sequences, approaches have been developed, that consider the 3D structure of proteins. A range of methods has sampled protein structures to regular volumetric 3D representations and assessed the quality of the structure (Derevyanko et al., 2018), classified proteins in enzymes classes (Amidi et al., 2017), predicted the protein-ligand binding affinity (Ragoza et al., 2017) and the binding site (Jiménez et al., 2017), as well as the contact region between two proteins (Townshend et al., 2019). While this is attractive, as 3D grids allow for unleashing the benefits of all approaches developed for 2D images, such as pooling and multi-resolution techniques, unfortunately, grids do not scale well to fine structures or many atoms, and even more importantly, they do not consider the primary and secondary structure of proteins.

Another approach that makes use of a protein's 3D structure, is representing proteins as graphs and applying GCNNs (Kipf & Welling, 2017; Hamilton et al., 2017). Works based on this technique represent each amino acid as a node in the graph, while edges between them are created if they are at a certain Euclidean distance. This approach has been successfully applied to different problems. Classification of protein graphs into enzymes, for example, have become part of the standard data sets used to compare GCNN architectures (Gao & Ji, 2019; Ying et al., 2018). Moreover, other works with similar architectures have predicted protein interfaces (Fout et al., 2017), or protein structure quality (Baldassarre et al., 2020). However, GCNN approaches suffer from over-smoothing, i. e., indistinguishable node representations after stacking several layers, which limits the maximum depth usable for such architectures (Cai & Wang, 2020).

It is also worth noticing that some of the aforementioned works on GCNN have considered different levels of protein structures indirectly by providing secondary structure type or distance along the sequence as initial node or edge features. However, these are not part of the network architecture and can be blended out due to the over-smoothing problem of GCNN. On the other hand, a recent protein function prediction method proposed by Gligorijevic et al. (2019) uses Long-Short Term Memory cells (LSTM) to encode the primary structure and then apply GCNNs to capture the tertiary structure. Also, the recent work from Ingraham et al. (2019) proposes an amino acid encoder that can capture primary and tertiary structures in the context of protein generative models. Unfortunately, none of these previous methods can incorporate all structural protein levels within the network architecture.

## 3 MULTI-GRAPH PROTEIN REPRESENTATION

To simultaneously take into account the primary, secondary, and tertiary protein structure during learning, we propose to represent proteins as a multi-graph $\mathcal{G} = (\mathcal{N}, F, \mathcal{A}, \mathcal{B})$. In this graph, atoms are represented as nodes associated with their 3D coordinates, $\mathcal{N} \in \mathbb{R}^{n \times 3}$, and associated features, $\mathcal{F} \in \mathbb{R}^{n \times t}$, $n$ being the number of atoms, and $t$ the number of features. Moreover, $\mathcal{A} \in \mathbb{R}^{n \times n}$ and $\mathcal{B} \in \mathbb{R}^{n \times n}$ are two different adjacency matrices representing the connectivity of the graph. Elements of matrix $\mathcal{A}$ are defined as $\mathcal{A}_{ij} = 1$ if there is a covalent bond between atom $i$ and atom $j$, and $\mathcal{A}_{ij} = 0$ otherwise. Similarly, the elements of matrix $\mathcal{B}$ are defined as $\mathcal{B}_{ij} = 1$ if there is a covalent or hydrogen bond between atom $i$ and atom $j$, and $\mathcal{B}_{ij} = 0$ otherwise.

### 3.1 INTRINSIC-EXTRINSIC DISTANCES

Differential geometry of surfaces (Pogorelov, 1973) defines intrinsic geometric properties as those, that are invariant under isometric mappings, i.e., under deformations preserving the length of curves on a surface. On the other hand, extrinsic geometric properties are dependent on the embedding of the surfaces into the Euclidean space. Analogously, in our protein multi-graph, we define intrinsic geometric properties as those that are invariant under deformations preserving the length of paths along the graph, i.e., deformations that preserve the connectivity of the protein. Additionally, we define extrinsic geometric properties as those, that depend on the embedding of the protein into the Euclidean space, i.e., on the 3D protein conformation. Using this terminology, we define three distances in

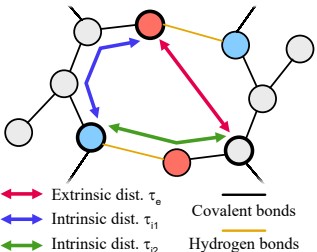

Figure 2: Distances between atoms in our protein graph.

our multi-graph, one extrinsic and two intrinsic (see Fig. 2). The extrinsic distance $\tau_e$ is defined by the protein conformation in Euclidean space, therefore we use the Euclidean distance between atoms, which enables us to capture the tertiary and quaternary structures of the protein. The intrinsic distances are inherent of the protein and independent of the actual 3D conformation. For the first intrinsic distance $\tau_{i1}$ we use the shortest path between two atoms along the adjacency matrix $\mathcal{A}$ of the graph, capturing the primary structure. The second intrinsic distance $\tau_{i2}$ is defined as the shortest path between two atoms along the adjacency matrix $\mathcal{B}$, capturing thus the secondary structure.

## 4 INTRINSIC-EXTRINSIC PROTEIN CONVOLUTION

The key idea of our work is to take into account the multiple invariances described in Sec. 1 during learning. Therefore, based on the success of convolutional neural networks for images (Krizhevsky

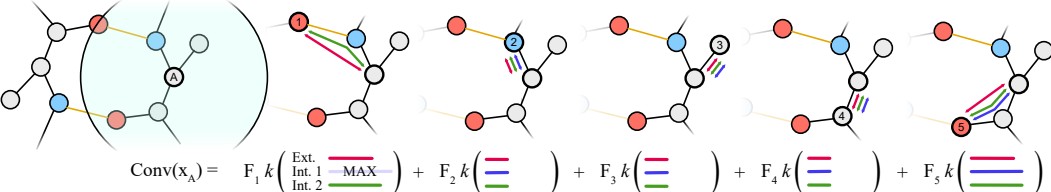

Figure 3: Intrinsic-extrinsic convolution on our multi-graph for atom A. First, we detect the neighboring atoms involved in the convolution using a ball query (a function returning all nodes closer than $r$ to a point $\mathbf{x}$). For each atom, the extrinsic (pink) and two intrinsic (blue and green) distances are input into the kernel and the result is multiplied by the atom's features. Lastly, all contributions from neighboring atoms are summed up.

et al., 2012) and point clouds (Qi et al., 2017a), in this paper we define a convolution operator for proteins which is able to capture these invariances effectively. To this end, we propose a convolution on 3D protein structures, which is inspired by conventional convolutions as used to learn on structured images. First, we define the neighborhood of an atom as all atoms at a Euclidean distance smaller than $m_e$. Moreover, we define our convolution kernels as a single Multi Layer Perceptron (MLP) which takes as input the three distances defined in Sec. 3.1, one extrinsic and two intrinsic, and outputs the values of all kernels. This enables the convolution to learn kernels based on one or multiple structural levels of the protein. Thus, the proposed convolution operator is defined as:

$$(\kappa * F)_k(\mathbf{x}) = \sum_{i \in \mathcal{N}(\mathbf{x})} \sum_{j=1}^{t} F_{i,j} \cdot \kappa_{j,k} \left( \frac{\tau_e(\mathbf{x}, \mathbf{x}_i)}{m_e}, \frac{\tau_{i1}(\mathbf{x}, \mathbf{x}_i)}{m_1}, \frac{\tau_{i2}(\mathbf{x}, \mathbf{x}_i)}{m_2} \right) \tag{1}$$

where $\mathcal{N}(\mathbf{x})$ are the atoms at Euclidean distance $d < m_e$ from $\mathbf{x}$, $F_{i,j}$ is the input feature $j$ of atom $\mathbf{x}_i$, $\kappa_{j,k}$ is the aforementioned kernel that maps $\mathbb{R}^3 \to \mathbb{R}$, $\tau_e(\mathbf{x}, \mathbf{x}_i)$ is the Euclidean distance between atom $\mathbf{x}$ and atom $\mathbf{x}_i$, $\tau_{i1}$ and $\tau_{i2}$ are the two intrinsic distances, and $m_e$, $m_1$, and $m_2$ are the maximum distances allowed ($m_1 = m_2 = 6$ hops in all experiments while $m_e$ is layer-dependent). All normalized distances are clamped to $[0, 1]$. The convolution, performed independently for every atom, is illustrated on a model protein in Fig. 3, where the distances between neighboring atoms are shown. This operation has all properties of the standard convolution, locality and translational invariance, and at the same time rotational invariant.

While we state our operation as an unstructured convolution as done in 3D point cloud learning (Hermosilla et al., 2018; Groh et al., 2018; Wu et al., 2019; Xiong et al., 2019; Thomas et al., 2019), it can also be understood in a message passing framework (Kipf & Welling, 2017) where hidden states are atoms, edges are formed based on nearby atoms and messages between atoms. The key difference to standard GCNNs is that we have edges for multiple kinds of bonds, use a hierarchy of graphs, and that the message passing function is learned.

## 5 Hierarchical Protein Pooling

We consider proteins at atomic resolutions. This allows us to identify the spatial configuration of the amino acid side chains, something of key importance for active site detection. However, proteins can have a large number of atoms, preventing the usage of large receptive fields in our convolutions (computational restriction) and limiting the number of learned features per atom (memory restriction).

Pooling is a technique commonly used in convolutional neural networks, as it hierarchically reduces the dimensionality of the data by aggregating local information (Krizhevsky et al., 2012). While it is able to overcome computation and memory restrictions when learning on images, unfortunately, it cannot be directly applied to proteins, as conventional pooling methods rely on discrete sampling. While, on the other hand, some of the techniques developed for unstructured point cloud data (Qi et al., 2017b; Hermosilla et al., 2018) could be applied to learn on the 3D coordinates of atoms, it is unclear what the edges of that new graph representation would be.

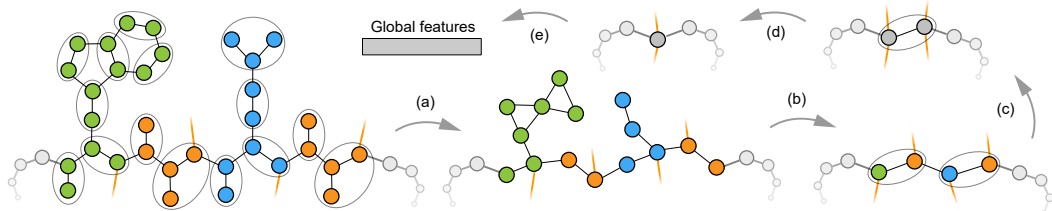

Figure 4: Hierarchical protein pooling: We segment the protein into amino acids (blue, orange, green). First, **(a)**, we apply spectral clustering on each independent amino acid graph. Then, **(b)**, each resulting amino acid is pooled to its $\alpha$-carbon. After that, we apply two backbone pooling operations, **(c)** and **(d)**. Lastly, we apply the symmetric operation average, **(e)**, to obtain the final feature vector.

Therefore, we define a set of operations that successively reduce the number of nodes in our protein graph. First, we iteratively reduce the number of atoms per amino acid to its alpha carbon. Afterward, we reduce the number of nodes along the backbone. Fig. 4 illustrates this process, which we describe in detail in the following paragraphs.

**Amino acid pooling** Simplified representations of amino acids have been previously used in Molecular Dynamics (MD) simulations, in order to get the number of computations down to a manageable level. Nevertheless, in contrast to our approach, these techniques do not use a uniform pooling pattern. Rather, while the atoms belonging to the backbone are not simplified, most of these approaches (Simons et al., 1997) simplify all atoms of the side chains to a single node. Other more conservative approaches, such as PRIMO (Kar et al., 2013) or Klein's models (DeVane et al., 2009), represent side chains with a variable number of nodes using a lookup table. However, they do not perform a uniform simplification, resulting in a high variance in the number of atoms per cluster. Moreover, these methods need to manually update the lookup table to incorporate rare amino acids.

In this work, we decided instead to follow the common practice in CNN for images where a uniform pooling is used over the whole image. We propose a method that is able to reduce the number of nodes in the protein graph by half. To this end, we generate an independent graph for each amino acid using the covalent bonds as edges and apply spectral clustering (von Luxburg, 2007) to reduce by half the number of nodes. Since the number of amino acids is finite (20 appearing in the genetic code), these pooling matrices are reused among all proteins (see Fig. 4 (a)). However, since the method only requires a graph as input, it can be directly applied to synthetic or rare amino acids.

From the amino acid pooling matrices, we create a protein pooling matrix $\mathcal{P} \in \mathbb{R}^{n \times m}$, where $n$ is the number of input atoms in the protein and $m$ is the number of resulting clusters in the simplified protein. This matrix is defined as $\mathcal{P}_{ij} = 1$ if the atom $i$ collapses into cluster $j$, and $\mathcal{P}_{ij} = 0$ otherwise. Using $\mathcal{P}$ we can create a simplified protein graph, $\mathcal{G}' = (\mathcal{N}', F', \mathcal{A}', \mathcal{B}')$, with the following equations:

$$\mathcal{N}' = \mathcal{D}^{-1}\mathcal{P}\mathcal{N} \qquad F' = \mathcal{D}^{-1}\mathcal{P}F \qquad \mathcal{A}' = \mathcal{P}\mathcal{A}\mathcal{P}^T \qquad \mathcal{B}' = \mathcal{P}\mathcal{B}\mathcal{P}^T \qquad (2)$$

where $\mathcal{D} \in \mathbb{R}^{m \times m}$ is a diagonal matrix with $\mathcal{D}_{jj} = \sum_{i=0}^{n} \mathcal{P}_{ij}$. Note, that the resulting matrices $\mathcal{A}'$ and $\mathcal{B}'$ might not be binary adjacency matrices, i. e., non-zero diagonal values or values greater than one. Therefore, we assign zeros to the diagonals and clamp edge values to one. These matrices are computed using sparse representations in order to reduce the memory footprint.

**Alpha carbon pooling** In a second pooling step, we simplify the protein graph to a backbone representation, whereby we cluster all nodes from the same amino acid to a single node. We define $\mathcal{P}$ accordingly, and the number of clusters $m$ is equal to the number of amino acids, while $\mathcal{P}_{ij} = 1$ if node $i$ belongs to amino acid $j$, and $\mathcal{P}_{ij} = 0$ otherwise. We use Eq. 2 to compute $F'$, $\mathcal{A}'$, and $\mathcal{B}'$. However, $\mathcal{N}'$ is defined as the alpha carbon positions of each amino acid since they better represent the backbone and the secondary structures.

**Backbone pooling** The last pooling steps are simplifications of the backbone chain. In each pooling step, every two consecutive amino acids in the chain are clustered together, effectively reducing by half the number of amino acids. Therefore, to compute $\mathcal{N}'$, $F'$, $\mathcal{A}'$, and $\mathcal{B}'$, we define $\mathcal{P}$ accordingly and use Equation 2. For single-chain proteins, for example, $\mathcal{P}_{ij} = 1$ if $\lfloor i/2 \rfloor = j$, or 0 otherwise.

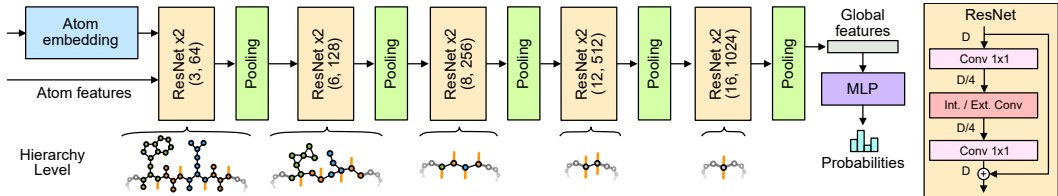

Figure 5: The architecture of our model. The input is composed of atom features and an atom embedding learned together with the network. Each layer is composed of two ResNet (He et al., 2016) bottleneck blocks, for which the radius in angstrom and the number of features are indicated in parentheses, followed by a pooling operation. An illustration of a single ResNet bottleneck block is presented in the right for $D$ input features (before each convolution we use batch normalization and a Leaky ReLU). The global protein features are processed by an MLP which computes the final probabilities. Protein graphs used in each level are indicated at the bottom.

## 6 EVALUATION

To evaluate our proposed protein learning approach, we specify a deep architecture (Sec. 6.1) which we compare to SOTA methods (Sec. 6.2) as well as different ablations of our approach (Sec. 6.3). During the entire evaluation, we focus on two downstream tasks in Sec. 6.4 and Sec. 6.5.

### 6.1 OUR ARCHITECTURE

By facilitating protein convolution and pooling, we are able to realize a deep architecture that enables learning on complex structures. In particular, the proposed architecture encodes an input protein into a latent representation which is later used in different downstream tasks.

The input of our network is a protein graph at atom level with a 6D input feature vector per each atom. The atom features comprise 1) covalent radius, 2) van der Waals radius, 3) atom mass, and 4-6) are the features of an atom type embedding learned together with the network. The input is then processed by 5 layers which iteratively increases the number of features, each composed of two ResNet (He et al., 2016) bottleneck blocks followed by a pooling operation (see Fig. 5). The respective receptive fields are [3, 6, 8, 12, 16] angstroms (Å) using [64, 128, 256, 512, 1024] features. The size of the receptive fields has been chosen to include a reduced number of nodes in it. As several proteins come without hydrogen bonds, we compute them using DSSP (Kabsch & Sander, 1983).

As the architecture is fully-convolutional, it can process proteins of arbitrary size, but after finitely many steps, it obtains an intermediate result of varying size. Hence, to reduce this to one final result vector, a symmetric aggregation operation, average, is used. Lastly, a single layer MLP with 1024 hidden neurons is used to predict the final probabilities.

### 6.2 OTHER ARCHITECTURES

We compare our architecture, as described above, to SOTA learners designed for similar downstream tasks. First, we compare to the latest sequence-based methods pre-trained unsupervised on millions of sequences: Rao et al. (2019); Bepler & Berger (2019); Alley et al. (2019); Strodthoff et al. (2020); Elnaggar et al. (2020). Second, we compare to different methods that take only the 3D protein structure into account: Kipf & Welling (2017); Diehl (2019); Derevyanko et al. (2018). Lastly, we also compare to two recent hybrid methods: Gligorijevic et al. (2019), who process primary structure with several LSTM cells first and then apply GCNN, and Baldassarre et al. (2020), who indirectly process primary and secondary structures by using as input distances along the backbone as edge features and secondary structure type as node features in a GCNN setup. When available, we provide the results reported in the original papers, while more details of the training procedures are provided in Appendix D.

Table 1: Comparison of our network to other methods on the two tasks (protein fold and enzyme catalytic reaction classification) measured as mean accuracy, where we outperform all methods.

| | Architecture | # params | FOLD | | | REACT |
|---|---|---|---|---|---|---|
| | | | Fold | Super. | Fam. | |
| HHSuite | | | 17.5 % | 69.2 % | 98.6 % | 82.6 % |
| Hou et al. (2018) | 1D CNN | 1.0 M | 40.9 % | 50.7 % | 76.2 % | |
| | 1D ResNet | 41.7 M | 17.0 % | 31.0 % | 77.0 % | 70.9 % |
| Rao et al. (2019)* | LSTM | 43.0 M | 26.0 % | 43.0 % | 92.0 % | 79.9 % |
| | Transformer | 38.4 M | 21.0 % | 34.0 % | 88.0 % | 69.8 % |
| Bepler & Berger (2019)* | LSTM | 31.7 M | 17.0 % | 20.0 % | 79.0 % | 74.3 % |
| Bepler & Berger (2019)† | LSTM | 31.7 M | 36.6 % | 62.7 % | 95.2 % | 66.7 % |
| Alley et al. (2019)* | mLSTM | 18.2 M | 23.0 % | 38.0 % | 87.0 % | 72.9 % |
| Strodthoff et al. (2020)* | LSTM | 22.7 M | 14.9 % | 21.5 % | 83.6 % | 73.9 % |
| Elnaggar et al. (2020)* | Transformer | 420.0 M | 26.6 % | 55.8 % | 97.6 % | 72.2 % |
| Kipf & Welling (2017) | GCNN | 1.0 M | 16.8 % | 21.3 % | 82.8 % | 67.3 % |
| Diehl (2019) | GCNN | 1.0 M | 12.9 % | 16.3 % | 72.5 % | 57.9 % |
| Derevyanko et al. (2018) | 3D CNN | 6.0 M | 31.6 % | 45.4 % | 92.5 % | 78.8 % |
| Gligorijevic et al. (2019)* | LSTM+GCNN | 6.2 M | 15.3 % | 20.6 % | 73.2 % | 63.3 % |
| Baldassarre et al. (2020) | GCNN | 1.3 M | 23.7 % | 32.5 % | 84.4 % | 60.8 % |
| Ours | | 9.8 M | **45.0** % | **69.7** % | **98.9** % | **87.2** % |

*Pre-trained unsupervised on 10-31 million protein sequences.

†Pre-trained on several supervised tasks with structural information.

## 6.3 ABLATIONS OF OUR METHOD

We study four axes of ablation: *convolution*, *neighborhood*, *pooling*, and *representation*. When moving along one axis, all other axes are fixed to `Ours` (●).

**Convolution ablation.** We consider four different ablations of convolutions: `GCNN` ( ) (Kipf & Welling, 2017); `ExConv` (●), kernels defined in 3D space only (Hermosilla et al., 2018); `InConvC` (●) uses only intrinsic distance $\tau_{i1}$; `InConvH` (●) uses only intrinsic distance $\tau_{i2}$; `InConvCH` (●) makes use of both intrinsic distances at the same time; `Ours3DCH` (●) uses both intrinsic distances plus distances along the three spatial dimensions; and lastly, `Ours` (●), that refers to our proposed convolution which uses both geodesics plus Euclidean distance.

**Neighborhood ablation.** We compare several methods to define our receptive field: `CovNeigh` (●), which uses the intrinsic distance $\tau_{i1}$ on the graph; `HyNeigh` (●); which uses the intrinsic distance $\tau_{i2}$, as well as `Ours` (●) which uses the Euclidean distance.

**Pooling ablation.** We consider six options: `NoPool` (●), which does not use any pooling operation; `GridPool` (●) overlays the protein with increasingly coarse grids and pools all atoms into one cell (Thomas et al., 2019); `TopKPool` (●) learns a per-node importance score which is used to drop nodes (Gao & Ji, 2019); `EdgePool` (●), which learns a per-edge importance score to collapse edges (Diehl, 2019); `RosettaCEN` (●), which uses the centroid simplification method used by the Rosetta software in Molecular Dynamics simulation (Simons et al., 1997); and our pooling, `Ours` (●).

**Representation ablation.** Lastly, we evaluate the granularity of the input, considering the protein at the amino acid level, `AminoGraph` (●), or at atomic level, `Ours` (●).

## 6.4 TASK 1: FOLD CLASSIFICATION (FOLD)

**Task.** Protein fold classification is of key importance in the study of the relationship between protein structure and function, and protein evolution. The different fold classes group proteins with similar secondary structure compositions, orientations, and connection orders. In this task, given a protein, we predict the fold class, whereby the performance is measured as mean accuracy.

Table 2: Study of ablations (rows) for the FOLD and REACTION tasks (columns).

| | | FOLD | | | REACT |
|---|---|---|---|---|---|
| | | Fold | Super. | Fam. | |
| Conv. | GCNN | 25.7 % | 46.5 % | 95.9 % | 84.9 % |
| | ExConv | 30.1 % | 46.3 % | 92.0 % | 85.0 % |
| | InConvC | 37.6 % | 65.1 % | 98.7 % | 85.4 % |
| | InConvH | 40.8 % | 62.0 % | 98.4 % | 85.5 % |
| | InConvCH | 43.5 % | 66.7 % | 98.7 % | 85.2 % |
| | Ours3DCH | 40.7 % | 62.2 % | 98.1 % | 85.8 % |
| | ●Ours | **45.0** % | **69.7** % | **98.9** % | **87.2** % |
| Neighbors | CovNeigh | 27.2 % | 41.5 % | 92.3 % | 41.6 % |
| | HyNeigh | 33.3 % | 50.6 % | 96.9 % | 56.9 % |
| | ●Ours | **45.0** % | **69.7** % | **98.9** % | **87.2** % |
| Pool | NoPool | 37.1 % | 59.8 % | 97.6 % | 84.7 % |
| | GridPool | 28.6 % | 41.8 % | 91.8 % | 86.1 % |
| | TopKPool | 40.7 % | 65.4 % | 98.4 % | 84.5 % |
| | EdgePool | 44.4 % | 69.6 % | **99.0** % | 86.9 % |
| | RosettaCEN | 41.7 % | 66.5 % | 98.9 % | 86.5 % |
| | ●Ours | **45.0** % | **69.7** % | 98.9 % | **87.2** % |
| Repr. | AminoGraph | 39.6 % | 64.7 % | **99.1** % | 85.3 % |
| | ●Ours | **45.0** % | **69.7** % | 98.9 % | **87.2** % |

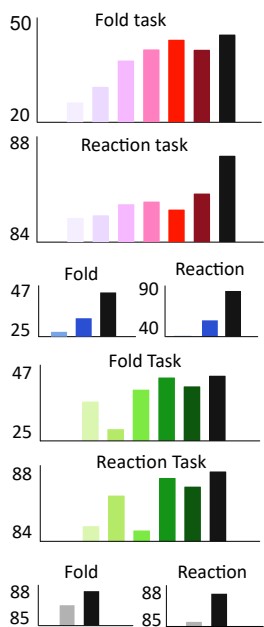

**Data set.** We use the training/validation/test splits of the SCOPe 1.75 data set of Hou et al. (2018). This data set consolidated $16,712$ proteins from $1,195$ folds. We obtained the 3D structures of the proteins from the SCOPe 1.75 database (Murzin et al., 1955). The data set provides three different test sets: Fold, in which proteins from the same superfamily are not present during training; Superfamily, in which proteins from the same family are not seen during training; and Family, in which proteins of the same family are present during training.

**Results.** Results, as compared to other methods, are reported in Tbl. 1. We find our method to perform better by a large margin without using additional features as input or pre-trained on additional data. Tbl. 2 shows how the tested ablations perform on this task. Overall we see that Ours (●) performs better than any ablation, indicating that our protein-specific representation, convolution, and pooling are all important ingredients. Lastly, we compare in the top of Tbl. 1 to the state of the art results reported in the paper (Hou et al., 2018) as to the results obtained using multiple sequence alignment, where we outperform all.

### 6.5 TASK 2: ENZYME-CATALYZED REACTION CLASSIFICATION (REACT)

**Task.** For this task, we classify proteins based on the enzyme-catalyzed reaction according to all four levels of the Enzyme Commission (EC) number (Webb, 1992). The performance is again evaluated as mean accuracy.

**Data set.** We collected a total of $37,428$ proteins from $384$ EC numbers. The data was then split into $29,215$ instances for training, $2,562$ instances for validation, and $5,651$ for testing. Note that all proteins have less than $50\%$ sequence-similarity in-between splits. A full description of the data set is provided in Appendix C.

**Results.** Again, our method outperforms previous works on this task (Tbl. 1). Ablations of our method (Tbl. 2) give further indication that all our components are relevant also for other tasks.

### 6.6 TASK 3: ONE-SHOT FOLD CLASSIFICATION (ONESHOT)

**Task.** The objective of this task is to give the binary answer to the question if a pair of proteins, which are both not observed during training, are in the same fold or not. This is also known as one-shot learning and we adopt the siamese architecture from Koch et al. (2015) using our protein encoder.

**Data.** We used the same dataset as for classification (Sec. 6.4), but withheld a random subset of 50 folds out of the original 1195 folds during training. During testing, we compare all proteins contained in the test set to all proteins contained in a reference set that comprises of all proteins of the original training set, and thus represents all 1195 folds. From each test protein, we select the pair with higher probability to predict the fold. To analyze how accuracy is affected by folds unseen during training, we divided the Fold test set from Task 1 in two different test sets. Our *seen* test set contains 419 proteins from 86 folds seen during training, while our *unseen* test set contains 299 proteins from the 50 folds not seen during training.

**Results.** When analyzing how many proteins could be predicted correctly, we obtained an accuracy of 39.0% for *seen* and an accuracy of 31.9% for *unseen*. This indicates that our method is able to generalize to folds that have not been seen during training. It also shows that our method is flexible enough to take concepts like the one from Koch et al. (2015), which uses hierarchical image convolutions, and generalize them to proteins. To view the resulting accuracies in the context of the other values reported in our paper, we also have trained the architecture for standard classification of Sec. 6.4 with the training set of this task composed of 1145 folds. We obtained an accuracy of 46.5% when testing against the 86 folds compared to the 39 % of the one-shot training.

## 7 CONCLUSIONS

Based on the multi-level structure of proteins, we have proposed a neural network architecture to process protein structures. The presented architecture takes advantage of primary, secondary, and tertiary protein structures to learn a convolution operator that works with intrinsic and extrinsic distances as input. Moreover, we have presented a set of pooling operations that enable the dimensionality reduction of the input mimicking the designs used by convolutional neural networks on images. Lastly, our evaluation has shown that by incorporating all the structural levels of a protein in our designs we are able to outperform existing SOTA protein learning algorithms on two downstream tasks, protein fold and enzyme classification.

Despite the reported success achieved in protein learning, some limitations apply to our idea. The main one being, that it requires the 3D structure of each protein, whilst other methods only make use of the protein sequence. This, however, may be alleviated by the advances in protein structure determination (Callaway, 2020) and prediction (Senior et al., 2020). Moreover, like other commonly used approaches such as GCNN, our convolution operator is invariant to rotations but also to chirality changes. This could be solved by incorporating directional information into the kernel's input. We leave this improvement for future work.

## ACKNOWLEDGEMENTS

This work was partially funded by the Deutsche Forschungsgemeinschaft (DFG), grant RO 3408/2-1 (ProLint), the project TIN2017-88515-C2-1-R(GEN3DLIVE), from the Spanish Ministerio de Economía y Competitividad, and by 839 FEDER (EU) funds.

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

# A    ADDITIONAL EXPERIMENTS

## A.1    ENZYME-NO-ENZYME CLASSIFICATION

**Task.** This is a binary classification of a protein into either being an enzyme or not. Even though this task is rather simple as compared to our other tasks, we have included it, as it has been extensively used as a benchmark for graph classification methods (Code), where ir is referred as D&D, and allows a direct comparison with several methods.

**Data set.** We downloaded the 1,178 proteins of the data set defined by Dobson & Doig (2003) from the Protein Data Bank (PDB) (Berman et al., 2000). Due to the low number of proteins, the performance is measure using $k = 10$-fold cross-validation.

**Results.** We compared to the results reported for methods that did not use extra training data and they were trained supervised as our method. We found our method to outperform all recent algorithms.

Table 3: Results for the D&D data set (Dobson & Doig, 2003).

| | |
|---|---|
| Gao & Ji (2019) | 82.4 % |
| Ying et al. (2018) | 82.1 % |
| Zhao & Wang (2019) | 82.0 % |
| Nguyen et al. (2020) | 81.2 % |
| Zhang et al. (2019) | 81.0 % |
| Togninalli et al. (2019) | 79.7 % |
| Zhang et al. (2018) | 79.4 % |
| Ours | **85.5** % |

## A.2    LATENT SPACE VISUALIZATION

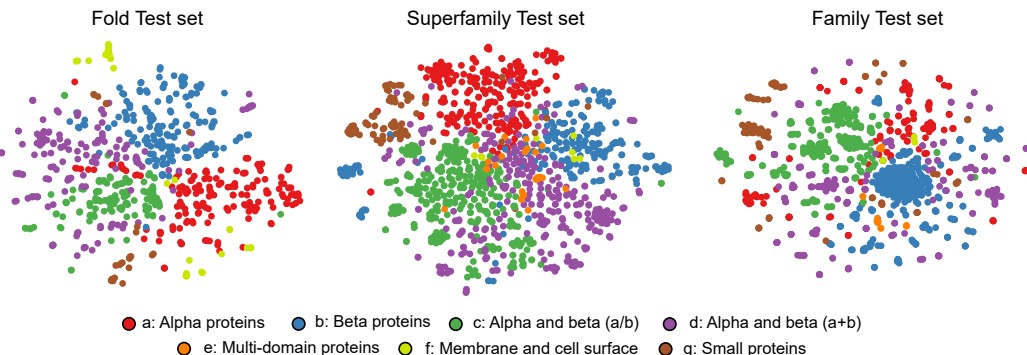

Figure 6: Visualization of the learned representations for the proteins of the different test sets of the SCOPe 1.75 data set. The data points are colored based on the higher level in the SCOPe hierarchy, showing that the network learned a latent space in which folds from the same class are clustered together.

In order to further validate that the network learned meaningful protein representations, we used t-SNE (van der Maaten & Hinton, 2008) to visualize the high-dimensional space of the representation learned for each protein on the different test sets of the SCOPe 1.75 data set (Hou et al., 2018), see Fig. 6. We use as protein representation the global feature vector before is fed to the final MLP. We color-coded each data point with the label of the higher level in the SCOPe hierarchy, protein class. We can see that, despite the network was not trained using this label, the network learned a representation in which folds are clustered by their upper class in the SCOPe hierarchy. In particular, the four major classes (a, b, c, and d) and the class of small proteins can be easily differentiate in the latent space. However, classes e and f are more difficult to identify. This could be due to the hyper-parameter tuning of t-SNE, or, in the case of class e, to the fact that proteins from this class are composed of more than one protein domain, each belonging to one of the other classes.

# B    AMINO ACID CLUSTERING MATRICES

Fig. 7 illustrates the clustering matrices generated by the spectral clustering algorithm (von Luxburg, 2007) for the 20 amino acids appearing in the genetic code.

# C    ENZYME REACTION DATA SET

Here we describe how the Enzyme Reaction data set was obtained and processed for the task described in Sec. 6.5. We downloaded EC annotations from the SIFTS database (Dana et al., 2018). This database contains EC annotations for entries on the Protein Data Bank (PDB) (Berman et al., 2000). From the annotated enzymes,

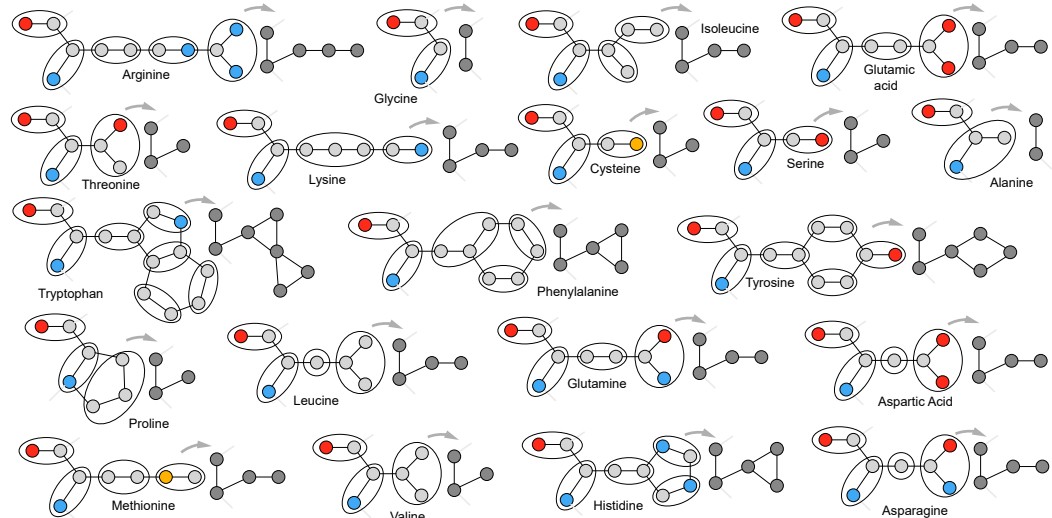

Figure 7: Visualization of the amino acid clustering matrices obtained with the spectral clustering algorithm (von Luxburg, 2007).

we cluster their protein chains using a $50\,\%$ sequence similarity threshold, as provided by PDB (Berman et al., 2000). We selected the unique complete EC numbers (e.g. EC 3.4.11.4) for what five or more unique clusters were annotated with it. For each cluster, we selected five different proteins (less than $100\,\%$ sequence similarity among them) as representatives. Note that even if the proteins are similar, selecting several chains per cluster can be understood as a type of data augmentation. In total, we collected $37,428$ annotated protein chains. Due to the low number of annotations for some of the enzyme types, the resulting data set contains a highly unbalanced number of protein chains per EC number.

Lastly, we split the data set into three sets, training, validation, and testing, ensuring that every EC number was represented in each set and all protein chains belonging to the same cluster are in the same set. This resulted in $29,215$ protein chains for training, $2,562$ protein chains for validation, and $5,651$ for testing.

## D    TRAINING AND HYPERPARAMETERS

### D.1    OUR ARCHITECTURE

We trained our model with Momentum optimizer with momentum equal to $0.98$ for $600$ epochs for the FOLD task and $300$ epochs for the REACTION. We used an initial learning rate of $0.001$, which was multiplied by $0.5$ every $50$ epochs with the allowed minimum learning rate of $1e-6$. Moreover, we used L2 regularization scaled by $0.001$ and we clipped the norm of the gradients to $10.0$. We used a batch size equal to eight for both tasks. We represented the convolution kernel with a single layer MLP with $16$ hidden neurons for the FOLD task and $32$ for the REACTION task.

To further regularize our model, we applied dropout with a probability of $0.2$ before each $1 \times 1$ convolution, and $0.5$ in the final MLP. Moreover, we set to zero all features of an atom before the Intrinsic / Extrinsic convolution with a probability of $0.05$ for the FOLD task and $0.0$ for the REACTION task. Lastly, we added Gaussian noise to the features before each convolution with a standard deviation of $0.025$.

In the FOLD task, we increased the data synthetically by applying data augmentation techniques. Before feeding the protein to the network, we applied random Gaussian noise to the atom coordinates with a standard deviation of $0.1$, randomly rotated the protein, and randomly scaled each axis in the range $[0.9, 1.1]$.

### D.2    RAO ET AL. 2019, BEPLER & BERGER 2019, ALLEY ET AL. 2019

For the FOLD task, we used the reported numbers in the paper (Rao et al., 2019) for the same data set. For the REACTION task, we downloaded the code and pre-trained models from the official repository of Rao et al. (2019) and pre-processed our data with their pipeline. We used the same training procedure as the one used by the authors: The weights of the net-

Table 4: Optimal batch size for the models from Rao et al. (2019), Bepler & Berger (2019), and Alley et al. (2019).

| Model | | Batch |
|---|---|---|
| | ResNet | 153 |
| (Rao et al., 2019) | LSTM | 221 |
| | Transf. | 90 |
| (Bepler & Berger, 2019) | | 240 |
| (Alley et al., 2019) | | 221 |

work were initialized with the values of a pre-trained model
on unsupervised tasks, and they were fine-tuned for the task.
We used a learning rate of $1e-4$ with a warm-up of 1000
steps. In order to find the optimal batch size for each model,
we trained each model increasing/decreasing the suggested
optimal batch size by multiplying/dividing by 2 until no improvement was achieved. Tbl. 4 presents the optimal
selected batch sizes.

### D.3 BEPLER & BERGER 2019 MULTITASK

We download the code and pre-trained models on the multitask method described in the paper of Bepler &
Berger (2019) from the official repository. Since this model was designed to predict feature embeddings for
amino acids, we added an attention-based pooling layer which computes a weighted aggregation of the amino
acid embeddings as a protein descriptor. This is then process by a 2 layer MLP to finally classify the protein.
This model was then trained by selecting the best hyper-parameters from different runs. We increased the batch
size by multiplying it by 2 and the best performance was achieved with a batch size of 64. The learning rate
was selected by multiplying it by 0.1, and the best performance was achieved by a learning rate of 0.001 for the
classification head and attention-based pooling layers and 0.00001 for the pre-trained model. During the first
epoch only the classification head and pooling layers where updated.

### D.4 STRODTHOFF ET AL. 2020

We downloaded the code and pre-trained models from the official repository (Strodthoff et al., 2020) and
pre-processed the data sets with their pre-processing pipeline. We used the same training procedure followed
by the authors, an initial learning rate of 0.001, which was decreased every time a new layer of the model was
unfreezed for fine-tuning. To evaluate the model, we averaged the performance of the forward and backward
pre-trained models. We experimented with different batch sizes and last layer sizes since in the original work the
last layers were composed of only 50 neurons. The best performance was obtained for a batch size of 32, and a
hidden layer of 1024 for the FOLD task and 512 for the REACTION task.

### D.5 ELNAGGAR ET AL. 2020

We downloaded the code and pre-trained model for the *ProtBert-BFD* network from the official repository (El-
naggar et al., 2020) and followed their code example for fine-tuning the network. We use the default parameters
for classification tasks and optimized the batch size by increasing and decreasing it multiplying by 2. The best
performance was achieved by a batch size of 32.

### D.6 KIPF & WELLING 2017

We implemented a GCNN architecture using the message passing defined by Kipf & Welling (2017). We
used the standard architecture of three GCNN which outputs are concatenated and give as input to a global
pooling ReadOut layer. The global features were processed by a single layer MLP with 1024 hidden layers. The
protein graph was defined using the contact map of the amino acid sequence with a threshold of 8 Å, as this
is common practice in GCNN literature (Gao & Ji, 2019). The initial amino acid features were learned with a
16D embedding layer that was trained together with the model. We trained the model until convergence with
momentum optimizer, a batch size of 8, and an initial learning rate of 0.001 which was multiplied by 0.5 every
50 epochs with an allowed minimum learning rate of $1e-6$. We also use dropout with probability equal to 0.5
in the final MLP and L2 regularization scaled by $5e-4$. All parameters were selected for optimal performance
using the same procedure as in our ablations.

### D.7 DIEHL 2019

The Diehl (2019) model was implemented by applying the pooling algorithm after each GCNN layer in the
model of Kipf & Welling (2017). The same hyperparameters were used to train this model.

### D.8 DEREVYANKO ET AL. 2018

We implemented the 3DCNN model of the paper Derevyanko et al. (2018) following the same number of layers,
features, and input resolution, 11 density volumes with a resolution of $120 \times 120 \times 120$. We trained the model
with momentum optimizer, a batch size of 16, and an initial learning rate of 0.005 which was multiplied by 0.5
every 50 epochs with an allowed minimum learning rate of $1e-6$. The model was trained until convergence.
For regularization, we used dropout layers with a probability of 0.5 in the final MLP and L2 regularization scaled
by $5e-4$. All parameters were selected for optimal performance using the same procedure as in our ablations.

### D.9   GLIGORIJEVIC ET AL. 2019

We downloaded the code from the official repository and pre-processed both data sets using their pipeline. We downloaded the pre-trained language model and run several experiments to select the best hyperparameters. We used a dropout rate of $0.5$, a learning rate of $2e - 4$, and a batch size of $16$. The model was trained until convergence.

### D.10   BALDASSARRE ET AL. 2020

We downloaded the code from the official repository and pre-processed both data sets using their pipeline. Since this method was designed to perform predictions at protein and amino acid level, we adapted the architecture by only using the global feature vector for the predictions and searching for the best architecture for our tasks. The best model had three GCNN layers, 128 out node features, 32 out edge features, and 512 out global features. We used a learning rate of $0.005$ scaled by $0.8$ every 50 epochs, weight decay of $1e - 5$, and batch size of $16$. The model was trained until convergence.

