# OpenReview forum: "Intrinsic-Extrinsic Convolution and Pooling for Learning on 3D Protein Structures"
_ICLR.cc/2021/Conference — ICLR 2021 Poster_

### Official Review · AnonReviewer4 · 2020-10-27
**Clearly described method achieving significant 3D protein structure gains**

**Rating:** 9
**Confidence:** 4

**Review:**

__Summary__
The authors describe a method to transform 3D protein structures for supervised machine learning. Their method introduces a convolution operation that considers both the intrinsic distances between atoms as defined by their bond structure and the extrinsic distances as defined by 3D proximity. They also introduce interpretable pooling operations developed using known biology of the amino acids. Overall, the method is effective and straightforward to follow due to having avoided unnecessary complexity. The figures greatly aid the reader.

The authors’ method outperforms a variety of competitive alternatives on protein fold and function classification tasks. These are important problems for which the authors’ model has achieved a significant performance boost. I don’t see why this model wouldn’t work well for any 3D protein structure labels that can be collected. They also perform a through ablation analysis to establish the contribution of the various components of their method.

__Major comments__
* I wasn’t able to understand what the “neighborhood” ablations represent and how they differ from “convolution” ablations. Are the neighbors used for anything other than the convolutions? For example, “CovNeigh” uses only the intrinsic distances, similarly to “InConvC”. What makes these different?

__Minor comments__
* On page 7, a Table 4 is mentioned that doesn’t appear to exist. I think they mean Table 3.

---

> ### Author Response · Authors · 2020-11-21
> **Ablation studies clarification**
>
> >What is the difference between the “CovNeigh” and the “InConvC” ablation.
>
> The ablation independently studies the two steps of the unstructured convolution:
> 1) to build a neighborhood $\mathbf{x} \in\mathcal N(x_i)$
> 2) loop over all atoms and map multiple distances to $\tau(\mathbf x, \mathbf x_i)$ to weights.
>
> “CovNeigh” would take (1) only those atoms that are linked by one type of chemical bound (covalent), and (2) perform as usual.
> “InConvC” would take (1) neighbours as usual, and (2) perform the kernel evaluation based on only one (the covalent) intrinsic distance.
> Essentially the first studies how the convolution input is built and the latter studies how it is weighted.
> ---
>
> >On page 7, a **Table 4** is mentioned that doesn’t appear to exist. I think they mean Table 3.
>
> Yes, this is a mistake. It now points to **Table 3**.

---

### Official Review · AnonReviewer1 · 2020-10-28
**Novel biologically-inspired ML with promising results**

**Rating:** 8
**Confidence:** 5

**Review:**

Pros:

- I think the paper is exceptionally well-written and the figures are very carefully designed. Applaud!

- Thank you for proper train/test/validation splits! Glad there are varying degrees of difficulty with proper held-out sequences.

- I very much appreciate proper comparison to other methods. Very thorough.

- Less important, but the model also performs better at these two tasks than any other approach. ( I say this because I believe the field shouldn't always require SoA if there is a significant technical advancement.)

Cons:

- The authors site "over-smoothing" for why their convolution operator performs better, but provide no direct evidence that this is the case. It needs to be noted that this is either a hypothesis, or more concrete evaluation of this needs to be performed to make this claim.

- Are there any replicates for standard error and ablation studies?

- Table 3 BLAST comparison is weak. JackHMMER or HMMER based tools are more appropriate than BLAST.

Neutral:

- What defines a hydrogen bond? This definition is clear to me in secondary structure, but seems more loose in tertiary structure.

- In your figures, it looks like only carbons, oxygens, and nitrogens are defined. What about hydrogens? If hydrogens aren't parameterized, how do you define hydrogen bonds? This may be good to clarify.

- In Table 2, does the modification of the architecture change the number of parameters?

- Definition of a "ball query" might be helpful.

- Are there any sequences with post-translational modifications in the dataset? If so, how are those handled?

---

> ### Author Response · Authors · 2020-11-21
> **Multiple sequence alignment and post-translational modifications**
>
> >What would be the evidence for “over-smoothing”?
>
> As we don’t have evidence for this, we have removed this statement.
>
> ---
>
> >Are there any “replicates for standard error and ablation studies”?
>
> Unfortunately we could not understand what the reviewer was referring to in this question.
>
> ---
>
> >_JackHMMER_ or _HMMER_ are more appropriate comparisons than _BLAST_.
>
> The revised paper now compares to _HHSuite_ in **Table 3**, which we understand to be the most recent and widely-accepted successor of _BLAST_.
> We see the conclusion that ours (45.0% / 69.7% / 98.9% for fold/superfold/family) performs better than multiple sequence alignment (17.5% / 69.2 % / 98.6%) unaffected.
>
> ---
>
> > Are hydrogens parameterized and how do you define hydrogen bonds?
>
> Hydrogens are not considered since most of the protein structures come without hydrogens (crystallography-based acquisition used in PDB does not have enough spatial resolution to capture hydrogens) and if they have them, they are usually added by computational methods. We compute hydrogen bonds using DSSP _Kabsch and Sander, 1983_.
>
> The revised paper now says “As several input proteins come without hydrogen bonds, they are computed in a pre-step using DSSP Kabsch and Sander (1983)”.
>
> ---
>
> >Does the modification of the architecture in **Table 2** change the number of parameters?
>
> Yes, more layers, more parameters. Also, a higher number of hidden layers in the kernel increases the number of parameters.
>
> ---
>
> >Can you define a "ball query"?
>
> The paper now says “.. ball query (a function returning all entities closer than $r$ to a point $x$)”.
>
> ---
>
> >How are post-translational modifications (PTMs) handled?
>
> PTM of amino acids and bound ligands or other molecules can provide cues of the function associated with the enzyme. Therefore, we only loaded atoms marked as “ATOM” in the PDB files and we ignored atoms marked as “HETATM”.
>
> ---
>
> >Are there any sequences with post-translational modifications (PTM) in the dataset? If so, how are those handled?
>
> Yes, our data contains PTM in the sense of proteins that contain one or more amino acids from the 15 non-standard amino acids defined by _Nagata et al_ [2014]. To understand their importance, we split the test set of 5651 proteins total into a set with (598) and without PTM (2491). When repeating testing of our method, we find that the accuracy is 81.9% for PTM and 87.4% for non-PTM. This indicates both are handled well.

---

### Official Review · AnonReviewer2 · 2020-10-28
**interesting architecture, but poorly framed, should be: multi-resolution graph MPNN**

**Rating:** 5
**Confidence:** 4

**Review:**

This paper proposes a graph neural network architecture that operates on the atoms in a protein structure.
It proposes a specific multigraph and pooling model structure, constructed using Euclidian distance and 3 types of edges (Euclidian, covalent, and hydrogen+cov).
There are three consecutive levels of granularity, with nodes corresponding to: (1) atoms (2) amino acids and (3) grouped amino acids.
The model is used to make a global prediction for a protein, and results are presented on the taks of Fold Classification and Reaction classification.

I recommend rejection for the paper in its current form, based on the concerns about the relevance of this method for fold classification (1/2 experiments), its framing as representation learning, and its framing as convolution vs graph neural networks.

## Strenghts:
* The two key model choices feel like a powerful choices for a graph neural network with an interesting domain-motivated set of architectural choices:
  - construction of the hypergraph with shortest-distance edges of 3 types
  - custom pooling of the graph from atom level nodes to groups of amino acids
* The paper has helpful visualizations and well-written, however see below for concerns around framing.
* Excellent ablation study in Table 2.

## Weaknesses:
* The key weakness is that the protein structure has to be provided as input to the network
    - ~~therefore fold classification is a flawed experiment as the full atomic coordinates is all that's needed for perfect assignment to the folds. Specifically comparisons to sequence-only classification (TAPE, Unirep, etc) are misleading.~~
    - as the authors point out, the amount of available data is tremendously less than sequence-only models. In fact the framing as "representation learning" is odd in this context, as there is no way to leverage unlabeled data, and no self-supervised objective is proposed.
* I find the framing of the method somewhat misleading, on a few counts:
    a) ~~representation learning~~ (see remarks abvove, no self-supervised + transfer of features)
    b) naming the core layer of the model a "convolution on 3D protein structures" is off. A crucial element of standard convolution is the regularity of the domain, while this is intrinsically graph structured data. Furthermore I believe the method still fits in the "neural message passing" framework (see bullet below). Therefore the proposed architecture seems to be much better summarized as "message passing graph neural network on a hierarchy of multi-graphs (hierarchy through protein pooling), with 3 types of graph edges defined by bonds and euclidian distances".
    c) after pooling, when vertices don't correspond to atoms but to clusters, the proposed convolution/GCN does not directly apply anymore. What are the edge connections at this stage?
* I disagree with the phrase "Although this operation could appear similar to message passing algorithms, they differ significantly". I believe the method fits in the MPNN framework, roughly as follows (notation following Gilmer et al 2017, renaming x, xi to v, w):
    - hidden state (per node) F_v with v the node (either atom, group of atoms, amino acid, cluster of amino acids depending on stage in the hierarchy)
    - edge introduced if euclidian distance d(vw) < m_e
    - edge features: 3 distances clamped [0,1]
    - learned message function $M = \kappa(e_{vw}) .  F_w$
    - $h_v^{t+1} = m_v^{t+1}$ - or possibly including batchnorm and relu.
The above re-formulation is quite close to GCN from Kipf & Welling (2016) but with more complex learned message function function of the 3 distances.

* Writing clarity: for eq (1) the notation needs to be introduced with dimensions (x, $\kappa$, F). Specifically for kernel $\kappa$ it needs to be made clear that $\kappa_j$ is a function from $R^3 \to R$ (?)
* Comments on experimental results:
    - ~~as mentioned above, I think fold classification is not an appropriate benchmark for this model~~
    - for enzyme reaction classification, a sota method on this problem should be included as benchmark. Ryu et al 2019 (DeepEC) seems relevant here, or a method based on HMM profiles.

-----
### Edit: reply to author's response and updated paper
(also see strikethroughs in the original review above)
* Fold classification: let me withdraw my concern here, and will defer to other reviewers & AC judgement if this task makes sense with protein structure as input -- indeed it may not to be a trivial task.
* Framing as (a) representation learning: improved in the updated paper, (b) convolution: still stands - the point cloud convs are not a very good comparison, since there is no graph structure there. (c) pooled coarsened graph stages: thanks for the pointer to end of Sec5.
* Positioning wrt message passing: the paragraph is a big improvement, removing some claims about over-smoothing. However re: "the message passing function is learned": this is still very much within the default MPNN framework from Gilmer et al. Altogether, the whole method would still be much better framed as a graph-based network, rather than shoehorning this into a description of a single "convolutional operator".  This will allow a proper discussion of what is currently the end of Sec5, where the graph does not correspond to an atom-level graph anymore, rather they now correspond to amino acid or coarser level graphs - it is confusing that this coarser graph stages are so briefly glossed over.
-- The citation to "can also be understood in a message passing framework (Kipf & Welling, 2017)" is off, should be "Gilmer et al., 2017" https://arxiv.org/abs/1704.01212

In conclusion, I am raising my score from 4->5, leaning towards 6. There is a lot of good work in this paper, and I would consider the paper a clear accept with the same method and same results, if it were thoroughly rewritten based on graph neural networks. Requiring full atomic structure as input to the method is the major limitation to the application and impact of the method.

---

> ### Author Response · Authors · 2020-11-21
> **Fold task and framing of the method**
>
> >Full atomic coordinates is all that's needed for perfect assignment to the folds.
>
> Not in practice.
>
> 1) Using the sequence is an interesting theoretical problem indeed, but results are slow and of limited quality. HHSuite, for example, has an accuracy of 17.5%, while ours has 45.0%.
>
> 2) Finding the right fold would require a database of all possible protein conformations and infinite time to search it exhaustively. In a finite world, this means that TM-align, a well-established package, requires 15 hours on six CPU cores to search 700 proteins against a reference set of 12k for an accuracy of 34%. We perform the same task in 16 seconds, trained on the same 12k proteins, at 45% accuracy, that is faster by three orders of magnitude.
>
> How pressing this problem is, can be seen from the fact that the widely used Protein Data Bank has just recently adopted a new algorithm (https://tinyurl.com/yxgt67ff) for 3D structural similarity.
>
> For similar reasons, one could also argue against using color in image classification: with it, things get easier. But using it right, if available, is an important problem to look into. In particular if it ultimately leads to better results.
>
> ---
>
> >Comparisons to sequence-only classification (_TAPE_, _Unirep_, etc) are misleading.
>
> We regret if this appears misleading to the reviewer, but we don’t see a way to avoid this: To the one hand, leaving them out, one could question if 3D structure has any benefit at all. To the other hand, including them could be perceived as unfair as their task is harder. We now explain in the paper: “These methods are baseline that only have access to the sequence. Including them documents the benefits of using 3D structure.”.
>
> Please note how we do not only outperform sequence-based methods, but also all methods using 3D structure.
>
> Please also note that there are several other 3D structure-based methods, that do not yet outperform sequence-based ones. To show this, the sequence-based baseline is required.
>
> ---
>
> >Should this be called “representation learning”?
>
> We have replaced “representation learning” with “protein learning”.
>
> ---
>
> > Is not calling this a "convolution on 3D protein structures" off, as it is not on a regular domain?
>
> There is published work we cite that refers to learned operations on unstructured 3D point clouds as “convolutions”
>
> https://arxiv.org/abs/1806.01759 (ACM Trans Graph 2018)
> https://arxiv.org/abs/1904.08889 (ICCV 2019)
>
> Our revised paper adds further citations along those lines
>
> https://arxiv.org/abs/1803.07289 (ACCV 2018)
> https://arxiv.org/abs/1811.07246 (CVPR 2019)
> https://arxiv.org/abs/1907.13079
>
> ---
>
> > Where do edges come from after pooling when nodes are not atoms anymore?
>
> New edges are created using Eq. 2 in the paper as it is stated in Sec. 5. The reviewer may also refer to Fig. 7 that visualizes old and new edges for all amino acids.
>
> ---
>
> > When authors claim their operation “appear similar to message passing algorithms”, but ”differs significantly", what is different?
>
> We should be specific in how they differ, indeed.
>
> The reviewer is correct to notice similarities: per-node state, edges to neighbors. We also believe the reviewer is right in noting the differences and novelty: learned message function, multiple metrics in one graph, using a hierarchy of graphs and pooling.
>
> Similarity and differences can be phrased as unstructured convolution or as an extension of graph convolutions. We don’t think it a weakness if a method can be explained in multiple frameworks.
>
> We would not agree that our approach is in any sense a special case of _Kipf and Welling 2016_ (no edge features, one type of edges, no hierarchy, no learned message passing, nothing protein specific).
>
> A revised paper now replaces the original statement by the following: “While we state our operation as an unstructured convolution (Hermosilla et al., 2018; Groh et al., 2018; Wu et al., 2019; Xiong et al., 2019; Thomas et al., 2019), it can also be understood in a message passing framework (Kipf & Welling, 2017) where hidden states are atoms, edges are formed based on spatial atom proximity and messages are passed between atoms. The key difference to existing work is, that we have edges for multiple kinds of bonds, use a hierarchy of graphs and that the message passing function is learned.”
>
> ---
>
> >Is the kernel in Eq. 1 a function from $\mathbb R^3$ to $\mathbb R$?
>
> Yes, the revised paper now gives the signature of $\kappa$.
>
> ---
>
> > Could you compare to _Ryu et al. [2019]_ (DeepEC)?
>
> Unfortunately, we are not able to manage this within the rebuttal phase.

---

### Official Review · AnonReviewer3 · 2020-10-29
**A sophisticated deep learning architecture for learning from protein structures, well motivated and well tested**

**Rating:** 9
**Confidence:** 4

**Review:**

This paper describes a deep learning architecture for representing and performing classifications on protein structures.  The representation involves three different distances: Euclidean distance and the shorted path between two atoms, where edges are either along covalent bonds or also include hydrogen bonds.  Each atom has a vector of associated features, and convolution is accomplished by defining a kernel on all three distances and then summing the features of each neighboring atom, weighted by the kernel value.  The paper also proposes three protein-specific pooling operations to cope with the large input size when representing all atoms in a protein.

Overall, this is an extremely clear paper, and the core ideas appear to be sound.  Furthermore, the experimental validation is quite extensive, and the results are impressively good.  Some positive points are that the authors consider several different tasks, and numerous state-of-the-art methods are included in the comparison.  I particularly appreciated the careful ablation study, demonstrating not just that the entire system works end-to-end but that the various pieces each contribute to its behavior.

The experimental setup appears to be valid. There is always the chance that these results could be optimistic due to (presumably unintentional) model selection happening during development of the proposed method, or because of a mismatch between the training data used for the published models and the test set used here.  But I can't see how the authors could have done a better job to guard against such issues, other than the obvious step of making their code and trained models publicly available.  It is unfortunate that the manuscript makes no mention of this.

One drawback to this work is its focus on recent literature.  I found it strange that the earliest citation in the related work section is from 2013.  The tasks being solved here have been the focus of extensive research going back 25 years or more.

The manuscript is up front about the fact that a drawback of the method is its requirement that the input proteins have known 3D structure.  However, another potential drawback is that the input does not take into account homology information drawn from, e.g., a sequence similarity search over a large protein database.  This information is typically represented as a PSSM column for each observed amino acid.  I would like to have seen this acknowledged, since it seems like a potentially valuable source of additional information.

A minor point: the introduction states that the model captures primary, seconary and tertiary structure, and then says that "As chain bindings affect the tertiary structure, the quaternary structure is captured implicitly."  But of course, this argument could apply to any of the other levels: amino acid sequence implicitly captures secondary and tertiary structure.

Incidentally, the Murzin cite has an incorrect year (1955).

---

> ### Author Response · Authors · 2020-11-21
> **Additional input and code availability**
>
> >Will the code and trained models be made publicly available?
>
> The code and training data is provided in the supplementary material already. Both will also be made public upon acceptance.
>
> ---
>
> >Would authors acknowledge how homology information, drawn from, e.g., a sequence similarity search over a large protein database is a valuable source of additional information?
>
> We did not use this information since we wanted to show that our method can learn end to end. However, this information could be easily incorporated by concatenating the features at the amino acid layer or at the atom level.
>
> ---
>
> >The claim that "As chain bindings affect the tertiary structure, the quaternary structure is captured implicitly." is a tautology and should be removed.
>
> We agree and have removed this statement.

---

### Official Review · AnonReviewer5 · 2020-11-10
**Interesting model architecture; but can the model generalize to unseen folds?**

**Rating:** 6
**Confidence:** 4

**Review:**

This paper presents new convolutional and pooling operators for protein structures. These components are used to design an architecture that shows strong performance on several downstream tasks.

The main strength of the paper is the presentation of new ideas for modeling protein structures. The proposed operators leverage the intuition behind convolutional networks but extend them for the protein case, e.g. by introducing rotational invariance in addition to translational invariance. The ideas themselves are interesting to machine learning researchers and useful to those working proteins. Due to the complexity of the model, I recommend that the authors release their code so that other researchers could evaluate these ideas on additional problems. The writing and presentation is clear.

Weaknesses:
- More updated baselines should be used. For example, for the sequence-only baselines, the authors should compare to ProTrans [Elnaggar, et al. 2020] or [Rives, et al. 2019] which show better results than the baselines used here. On the structural side, the authors should compare to the architectures proposed by [Du, et al. 2019] or [Anand, et al. 2020].
- A key sequence baseline is missing: multiple sequence alignments.
- The only tasks considered are classification tasks. The paper could be improved by evaluating on more practical tasks, such as protein design, e.g. the tasks in [Du, et al. 2019] or [Anand, et al. 2020]. The architecture described here could be very useful in those settings.
- The authors compare to Bepler, et al. (2019) which is a great baseline since it uses both sequence and structural information. However, it appears from the text that the authors used the version of this model provided by Rao, et al. However, Rao et al. simply used the architecture from Bepler, et al. and re-trained it on sequence data only. Therefore, I recommend that the authors retrieve the weights from Bepler, et al. directly.
- On the fold classification task, the hardest test set considered is "Fold, in which proteins from the same superfamily are not present during training." It would be interesting to evaluate the model on a harder generalization setting in which proteins from the same fold are also not present during training. The delta between this model and DeepSF decreases when the sets go from family -> superfamily -> fold. To complete the picture, it would be important to go one step further.
- Relatedly, the authors have not demonstrated that the models can generalize to novel folds. Without demonstrating this, the model cannot be used for important tasks, such as protein design. The paper would be much more compelling if the authors could show that their architecture generalizes better than prior work. To accomplish this, the authors would need to move beyond a classification framework toward a clustering framework because it's impossible for a classifier to predict novel folds.
- The names of the test splits on the fold classification task is non-standard. Generally, "fold split" means that proteins from the same fold are not included in the same set; "superfamily split" means that proteins from the same superfamily are not included in the same set, etc. What the authors call the family split ("in which proteins of the same family are present during training") is usually not included as overfitting to / memorizing the training set could still result in good performance (perhaps this is why the proposed model scores 98.9% here).

To summarize the weaknesses: more work is needed on the baselines and metrics. Additional evidence is also needed to support that the model can generalize to unseen folds.

Overall, this paper is a great start and the proposed model architecture could be interesting to ML researchers and practitioners in the biology space. In its current state, this is a borderline paper because it is missing a critical component of generalization of novel folds, which is necessary for this model to have significant impact in the field. If the authors can resolve my concerns during the rebuttal period, I am willing to raise my score.

Update: The authors have included an additional experiment around fold generation in Sec 6.6. However, no baselines are included, so it is difficult to understand the result in context and understand how this method generalizes compared to existing methods. The authors have also included two additional baselines: Bepler, et al. and MSAs. More analysis is needed to compare this with SOTA in representation learning. The authors compare to "Elnaggar et al. (2020)" but it isn't clear which model was used. Elnaggar et al. (2020) have released a series of different models. The authors should clarify this in the camera-ready and ensure they used the best models released by Elnaggar et al. I have increased my score.

---

> ### Author Response · Authors · 2020-11-21
> **Generalization to new folds and other experiments**
>
> >Will you release code?
>
> The code can be found in the submitted supplementary material. The revised paper now includes the (anonymized) Github URL.
>
> ---
> >Can it generalize to unseen folds?
>
> We interpret this question to ask for one-shot learning: during testing, a network is presented with a protein from a fold unseen during training. Without having learned any classes, the method has to return the probability that the input protein is in the same fold as a reference protein, also not seen during training.
> Following _Koch et al.’s_ popular “Siamese Neural Networks for One-shot Image Recognition” we have realized this challenging task with our architecture. During training, we withheld 50 out of the original 1195 folds. During testing, we compare all proteins contained in the test set to all proteins contained in a reference set that comprises out of all proteins, and thus represents all 1195 folds. From each test protein, we select the pair with higher probability to predict the fold. To analyze how accuracy is affected by folds unseen during training, we divided the Fold test set from Task 1 in two different test sets. Our _seen_ test set contains 400 proteins from 86 folds seen during training, while our _unseen_ test set contains 300 proteins from the 50 folds not seen during training. When analyzing how many proteins could be predicted correctly, we obtained an accuracy of 39.0% for _seen_ and an accuracy of 31.9% for _unseen_. This indicates that our method is able to generalize to folds that have not been seen during training. It also shows that our method is flexible enough to take concepts like the one from _Koch et al._, which uses hierarchical image convolutions, and generalize them to proteins. To view the obtained accuracies in the context of the other values reported in our paper, we also have trained the architecture for standard classification of Sec. 6.4 with the training set of this task, composed of  1145 folds. We obtained an accuracy of 46.5% when testing against the 86 folds compared to the 39% of the one-shot training.
> The paper has been revised to include these experiments in the result **Sec. 6.6**.
>
>
> ---
> > Would clustering be an option for discovering new folds?
>
> We did not yet explore the option of clustering in latent space. However, our convolutional architecture gives a good basis to build an encoder-decoder. Its latent space could be used in approaches like DEC by Xie et al. “Unsupervised Deep Embedding for Clustering Analysis” ICML 2016.
>
> ---
> >Could you compare to multiple sequence alignment?
>
>
> The revised paper now compares to _HHSuite_ in **Table 3**.
>
> We see the conclusion that ours (45.0% / 69.7% / 98.9% for fold/superfold/family) performs better than multiple sequence alignment (17.5% / 69.2 % / 98.6%) unaffected.
>
> ---
> >Should comparison to  _Bepler, et al_. be done using their weights, not the ones from _Rao et al._?
>
>
> We downloaded _Bepler et al_’s weights and added results to **Table 2**. The conclusions, including _Ours_ > _Bepler et al._ > _Rao et al._, are unaffected.
>
> ---
> >Could you compare to _ProTrans [Elnaggar, et al. 2020]_
>
> We have downloaded the code from the official repository and trained in our tasks. The revised version of the paper now includes these experiments.
>
> ---
> >Could you compare to _Rives, et al. [2019]_ and / or  _Du, et al. [2019]_?
>
> This comparison was too complex and lengthy and hence we weren't able to conduct it in the rebuttal phase.
>
> ---
> >Could you compare to _Anand, et al. [2020]_?
>
>
> Unfortunately, we could not find code or data of that method to compare to.
>
> ---
> >Are the names of the fold classification task standard?
>
> The convention is adopted from Sec. 2.1 and Section I.1.1 of _Hou et al 2018_’s “DeepSF: deep convolutional neural network for mapping protein sequences to folds”, who created the dataset. The same terminology was used by  _Rao et al._, “Evaluating Protein Transfer Learning with TAPE”.

---

### Author Response · Authors · 2020-11-21
**Thank you for the reviews**

First, we would like to thank all the reviewers for their valuable comments and suggestions. We have uploaded a revised version of the paper addressing most of their concerns and we answered each reviewer separately summarizing each comment into a sentence followed by the response.
Sincerely,
The authors

---

### Decision · Program_Chairs · 2021-01-07
**Final Decision**

**Decision:**

Accept (Poster)

**Comment:**

Protein molecule structure analysis is an important problem in biology that has recently become of increasing interest in the ML field. The paper proposes a new architecture using a new type of convolution and pooling both on Euclidean as well as intrinsic representations of the proteins, and applies it to several standard tasks in the field.

Overall the reviews were strong, with the reviewers commending the authors for an important result on the intersection of biology and ML. The reviewers raised the points of:
- weak baselines (The authors responded with adding suggested comparison, which were not completely satisfactory)
- focus mostly on recent protein literature
- the reliance of the method on the 3D structure. The AC however does not find this as a weakness, as there are multiple problems that rely on 3D structure, which with recent methods can be predicted computationally rather than experimentally.

We believe this to be an important paper and thus our recommendation is Accept. As the AC happens to have expertise in both 3D geometric ML and structural biology, he/she would strongly encourage the authors to better do their homework as there have been multiple recent works on convolutional operators on point clouds, as well as intrinsic representation-based ML methods for proteins.